# Flavin affinity for the reductase HpaC differentially sensitizes *Neisseria gonorrhoeae* during Type IV pilus-dependent killing

Linda I. Hu[1]*, Egon A. Ozer[2], H. S. Seifert[1]

1 Department of Microbiology-Immunology, Northwestern University Feinberg School of Medicine, Chicago, Illinois, United States of America, 2 Department of Medicine, Division of Infectious Diseases, Feinberg School of Medicine, Northwestern University, Chicago, Illinois, United States of America

* lihu@northwestern.edu

## Abstract

The *Neisseria gonorrhoeae* Type IV pilus is a dynamic fiber involved in host cell attachment, DNA transformation, twitching motility, and evading the innate immune system. We previously reported that pilus expression affects iron homeostasis and sensitivity to killing by oxidative (iron-dependent antibiotic streptonigrin and hydrogen peroxide and non-oxidative (antimicrobial peptide LL-37) agents. Here, we use *in vitro* evolution to identify genes involved in *N. gonorrhoeae* susceptibility to streptonigrin. We identified a mutation in the NGO0059 locus that encodes HpaC that results in a glycine to cysteine change in position 93. Although HpaC homologs are known as part of a two-component FAD-dependent monooxygenase system consisting of an *hpaC* reductase and an *hpaB* monooxygenase, *Neisseria* lack the monooxygenase. While HpaC increases streptonigrin sensitivity, HpaC also promotes hydrogen peroxide and LL-37 resistance. We tested whether the HpaC effect in streptonigrin, hydrogen peroxide and LL-37 sensitivity involved the Type IV pilus. We determined that HpaC affects streptonigrin independently of the pilus while hydrogen peroxide- and LL-37-mediated killing involves both HpaC and the pilus. We demonstrate that the Gly93Cys change conferred enhanced affinity for FAD and resulted in a loss-of-function phenotype in streptonigrin susceptibility. These data suggest that HpaC's role in FAD oxidation and reduction impacts pilus-dependent and -independent resistance against neutrophil-mediated killing.

## Author summary

*Neisseria gonorrhoeae* is the causative agent of the sexually transmitted disease, gonorrhea. Neutrophils are first host immune responders that activate multiple potent antimicrobial defences including reactive oxygen species and antimicrobial peptides. Yet, *N. gonorrhoeae* is highly resistant to these host

**Data availability statement:** The data that support the findings of this study are publicly available from Prism with the identifier DOI https://doi.org/10.18131/rv6c4-rdb68.

**Funding:** This work was supported by National Institute of Allergy and Infectious Diseases (https://www.niaid.nih.gov/) grants R01 AI146073 and R21 AI148981 to HSS. The funders had no role in study design, data collection and analysis, decision to publish, or preparation of the manuscript. The content is solely the responsibility of the authors and does not necessarily represent the official views of the National Institutes of Health. This manuscript is the result of funding in whole or in part by the National Institutes of Health (NIH). It is subject to the NIH Public Access Policy. Through acceptance of this federal funding, NIH has been given a right to make this manuscript publicly available in PubMed Central upon the Official Date of Publication, as defined by NIH.

**Competing interests:** The authors have declared that no competing interests exist.

protection mechanisms. The Type IV pilus is known to promote resistance to these antibacterial agents; however, how the pilus does this was not known. We show that an FAD reductase HpaC has pilus-dependent and pilus-independent functions that contribute to this bacterium's resistance through its interaction with flavin.

## Introduction

*Neisseria gonorrhoeae* (Gc, gonococcus) is the causative agent of gonorrhea, the second leading bacterial sexually transmitted infection behind chlamydia. Gc colonizes the urogenital tract and can cause epididymitis in men and pelvic inflammatory disease in women [1]. If left untreated, the infection can lead to complications of ectopic pregnancy, infertility, neonatal blindness, arthritis and, in extreme cases, sepsis [2–4].

As Gc do not express exotoxins, colonization drives pathology by increasing local inflammation. Mucosal epithelial cells and macrophage pattern recognition of shed peptidoglycan fragments and components of the outer membrane trigger neutrophil recruitment to the site of infection [5,6]. The Gc surface opacity proteins bind to the human neutrophil receptor CEACAM3 and activate a signaling cascade resulting in the further release of neutrophil recruiting cytokines [7]. This forward-feeding circuit continues, amplifying the inflammatory response. While neutrophil activation fights the infection, tissue damage also occurs. As a result, neutrophils play a key role in Gc pathogenesis.

Despite these waves of incoming neutrophils, they do not clear Gc infections. During an oxidative burst, the enzyme complex NADPH oxidase assembles on the neutrophil phagosomal and plasma membranes and converts oxygen to superoxide and secondary ROS, which includes hydrogen peroxide ($H_2O_2$). ROS oxidize and damage DNA, lipids, proteins and iron-sulfur clusters [8]. Neutrophils also release their genomic DNA in a process called NETosis that encases bacterial cells in a mesh. NETosis and degranulation release degradative enzymes like proteases and lysozyme and cationic antimicrobial peptide LL-37 [9] which target and perforate the bacterial membrane [10]. Nonetheless, neutrophil-rich secretions carry viable Gc that can survive within neutrophils due to Gc expressing factors that resist neutrophil antimicrobials [11–18], one of which is the Type IV pilus (T4p) [19]. The T4p is a dynamic fiber that can extend and retract through its assembly and disassembly. T4p is selected for in experimental infections [20,21] and is essential for multiple functions, including colonization, adherence to the host epithelium, twitching motility on surfaces [22], formation of cell aggregates with neighboring Gc [23], and transformation of extracellular DNA [24].

The most recent T4p function reported was promoting resistance to neutrophil-mediated killing [25]. In the absence of the gene *pilE* that encodes the major pilus subunit, pilin, Gc cells are sensitive to antimicrobial peptide LL-37 and $H_2O_2$ [25]. Using the bactericidal antibiotic streptonigrin (SPN), we determined that the

mechanism of pilus-mediated resistance to neutrophils involves iron homeostasis [19]. The soil bacterium *Streptomyces flocculus* produces SPN which binds to labile intracellular iron, interacts with DNA and locally generates reactive oxygen species [26]. Resistance to SPN, $H_2O_2$, and LL-37 was restored to the Δ*pilE* mutant when iron was limited [19]. Here, we used *in vitro* evolution to isolate nonpiliated Gc mutants with increased SPN resistance. This approach led us to discover a role for an FAD reductase HpaC in SPN resistance previously known for its role in aromatic compound metabolism.

## Results

### *In vitro* evolution of *pilE* mutant to increase SPN resistance

We previously reported that a Δ*pilE* mutant (N-1–69, Table 1) was more sensitive to SPN than the FA1090 parental strain N-1–60 [19]. To identify genes involved in SPN sensitivity, we performed *in vitro* evolution of a Gc *pilE* mutant to increase SPN resistance. We treated a mid-exponential phase Δ*pilE* Gc culture with 0.5 μM SPN for 30 minutes. This SPN treatment dose and duration results in the relative survival of between $10^{-2}$ to $10^{-3}$ Δ*pilE* cells compared to ≥$10^{-1}$ piliated,

**Table 1. Strains and plasmids.**

| Strain or plasmid | Description | Reference/ source |
|---|---|---|
| *Strains* | | |
| N-1–60 | FA1090 multisite G4 mutant 1–81-S2 pilE variant, pilC1PL | Hu and Yin et al 2020 |
| N-1–69 | An unmarked Δ*pilE* mutant (deletion from the 6th amino acid to the stop codon in *pilE* from Alison Criss) in N-1–60 | Hu and Yin et al 2020 |
| N-5–22 | CRISPRi-*pilE* in N-1–60 | Geslewitz et al 2023 |
| N-7–15 | *hpaC*(Gly93Cys) point mutation in N-5–22 | this study |
| N-7–37 | Δ*hpaC*::*kan* mutation in N-1–60 | this study |
| N-7–52 | Δ*pilE* (N-1–69) in N-7–37 | this study |
| N-8–53 | N-8–57 with IPTG-inducible *hpaC* cloned in the *lctP*/*aspC nics* chromosomal region | this study |
| N-8–57 | In-frame replacement of *hpaC* ORF with 18 bp EcoR1-EcoRV-BamH1 polylinker complemented with *hpaC* in the *nics* site, ErmR | this study |
| N-8–63 | N-8–65 with IPTG-inducible *hpaC* cloned in the *lctP*/*aspC nics* chromosomal region | this study |
| N-8–65 | unmarked Δ*pilE* mutant from N-1–69 in N-8–57 | this study |
| N-9–38 | N-8–57 with IPTG-inducible *hpaC(Gly93Cys)* cloned in the *lctP*/*aspC nics* chromosomal region | this study |
| N-9–1 | N-8–57 with IPTG-inducible *hpaC(Gly93Ala)* cloned in the *lctP*/*aspC nics* chromosomal region | this study |
| *Plasmids* | | |
| pET28a-FA1090-HpaC | pET28a plasmid carrying *hpaC* [NGO0059], KanR | this study |
| pET28a-FA1090-HpaC(Gly93Cys) | pET28a plasmid carrying *hpaC* with a G277-T277 mutation, KanR | this study |
| pGCC4 | IPTG-inducible Neisseria chromosomal complementation (nics) vector, KanR and ErmR | Mehr and Seifert 1998 |
| pGCC4-FA1090-HpaC | pGCC4 plasmid with *hpaC* cloned into the PacI/PmeI site, KanR and ErmR | this study |
| pGCC4-FA1090-HpaC(Gly93Cys) | pGCC4 plasmid with *hpaC(Gly93Cys)* cloned into the PacI/PmeI site, KanR and ErmR | this study |

PLOS Pathogens

parental cells. After six sequential rounds of growth and SPN selection, we tested 16 isolates to determine whether the population demonstrated an increased level of SPN resistance compared to the unevolved Δ*pilE* parental strain. We observed increased resistance compared to the control strain for all the isolates ([Fig 1A]). Whole genome sequencing and variant calling of two of these isolates showed eight genetic changes ([S1 Table]) compared to an unevolved strain. Six intragenic mutations (a missense mutation in NGO0059 *hpaC*, a synonymous mutation in NGO0650 *recN*, and phase variations in NGO11100 *opaB*, NGO09965 *opaE*, NGO1765 *pglA*, and NGO2158 *lgtD*) and two intergenic phase variations

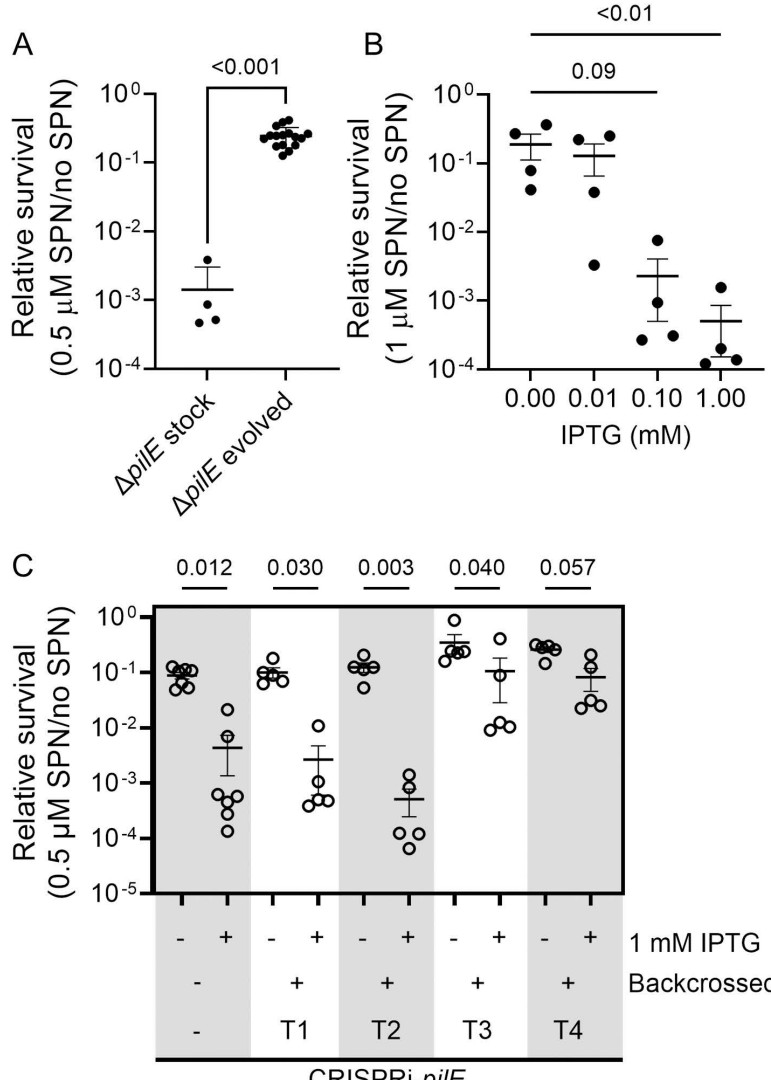

**Fig 1. Using *in vitro* evolution to increase SPN resistance in the absence of pilin.** (A) A comparison of SPN sensitivity in the unevolved Δ*pilE* mutant (N-1-69) to 16 clones after *in vitro* evolution with streptonigrin. Significance was determined using a Student's *t* test. Shown are the means and standard deviations. (B) Repressing *pilE* expression using an IPTG-inducible CRISPR interference system [27] reduces SPN resistance. The medians are shown for biological replicates. Statistical significance was determined using a one-way ordinary analysis of variance (ANOVA) followed by a Dunn's multiple comparisons test. (C) SPN relative survival of four CRISPRi-*pilE* strains isolated from a backcross (Transformant 1, 2, 3, and 4) using genomic DNA from the evolved, SPN resistant Δ*pilE* mutant compared to the control CRISPRi-*pilE* strain (N-5-22) in the presence and absence of *pilE* repression using IPTG. The means of biological replicates and the standard error are plotted.

(one between NGO2093 *fetA* and NGO02094 *groES* and another between NGO1810 hypothetical gene and NGO1811 *truA*).

## Isolating an *hpaC* point mutation that is responsible for SPN resistance

To discriminate between observed mutations that were responsible for SPN resistance and those unrelated to SPN sensitivity, we transformed the evolved strains' genomic DNA into a strain that has conditional *pilE* expression using IPTG-regulated CRISPR interference (CRISPRi-*pilE* strain N-5–22, Table 1) [27]. As expected, CRISPRi-*pilE* exhibited an IPTG concentration-dependent sensitivity to SPN (Fig 1B). We treated the transformed cells with SPN in the presence (*pilE* is repressed) and absence of IPTG (*pilE* is expressed). Two transformants (T1 and T2 in Fig 1C) showed similar relative survival as the untransformed parental strain. Interestingly, two transformants (T3 and T4 in Fig 1C) were less sensitive to SPN than transformants T1 and T2.

We compared the genomic sequence of T3 and T4 to T1 and T2 (Table 2) and identified six mutations. One mutation was a nonsynonymous point mutation identified within the *hpaC* gene (NGO0057) in the SPN-resistant isolates T3 and T4. A G-to-T substitution at nucleotide position 277 results in a Gly-to-Cys change at amino acid position 93 of HpaC (Table 2). Five mutations were intergenic or intragenic variations of polynucleotide repeat numbers previously identified to undergo stochastic phase variation [28–30]. However, these polynucleotides repeat variations were not specific to the resistant clones T3 and T4.

## HpaC promotes SPN sensitivity independently of the T4p

To determine the role of HpaC in SPN resistance, we replaced the *hpaC* open reading frame with a kanamycin resistance cassette in piliated N-1–60 and nonpiliated Δ*pilE* backgrounds. The *hpaC* insertional mutant (Fig 2A) had a similar SPN phenotype to the Gly93Cys mutation (Fig 2B): deleting *hpaC* promoted SPN resistance in both piliated and nonpiliated cells and the effect was greater in nonpiliated Gc. We also complemented the *hpaC* null mutant in piliated (*hpaC*/*nics*::*hpaC*, N-8–53, Table 1, Fig 3A) and nonpiliated Gc (Δ*pilE*Δ*hpaC*/*nics*::*hpaC*, N-8–63, Table 1, Fig 3B).

The similar SPN sensitivity patterns between the deletion of *hpaC* and the *hpaC(Gly93Cys)* mutant suggested that *hpaC(Gly93Cys)* is a loss-of-function mutation. Consistent with this assertion, overexpressing WT *hpaC* inhibited SPN resistance while overexpressing *hpaC(Gly93Cys)* had no effect (Fig 3C). This residue is not critical for HpaC-mediated inhibition of SPN sensitivity because an *hpaC(Gly93Ala)* mutant partially complemented SPN resistance (Fig 3C). These

**Table 2. Whole genome sequencing of SPN sensitive and resistant transformants.**

| Transformant | | | | | |
|---|---|---|---|---|---|
| T1$^S$ | T2$^S$ | T3$^R$ | T4$^R$ | Mutation | Gene [locus] |
| *hpaC(Gly93)* | *hpaC(Gly93)* | *hpaC(Cys93)* | *hpaC(Cys93)* | missense variant | *hpaC* [NGO0059] |
| (G)$_{14\to13}$ | (G)$_{14}$ | (G)$_{14}$ | (G)$_{14\to13}$ | phase variant | *lgtC* [NGO11620] |
| (G)$_{13}$ | (G)$_{13\to14}$ | (G)$_{13\to12}$ | (G)$_{13}$ | phase variant | *lgtD* [NGO2158] |
| (CGCAA)$_{14\to16}$ | (CGCAA)$_{14}$ | (CGCAA)$_{14}$ | (CGCAA)$_{14}$ | phase variant | hypothetical [NGO0366] |
| (C)$_{9\to12}$ | (C)$_{9\to10}$ | (C)$_{9\to12}$ | (C)$_{9\to12}$ | intergenic phase variant | *fetA* [NGO2093] - *groES* [NGO2094] |
| (C)$_{10}$ | (C)$_{10\to11}$ | (C)$_{10\to11}$ | (C)$_{10\to11}$ | intergenic phase variant | hypothetical [NGO1810] - *truA* [NGO1811] |

Table 2 shows the whole genome comparison of four backcrossed strains.

"S" and "R" indicate whether the transformant (T) was relatively more resistant (R) or not (S) to streptonigrin than the parent strain CRISPRi-*pilE* (N-5–22) when *pilE* expression is repressed.

The arrows indicate the change in repeat number compared to the parent strain CRISPRi-*pilE* (N-5–22). No arrow indicates that the sequence did not change compared to the CRISPRi-*pilE*.

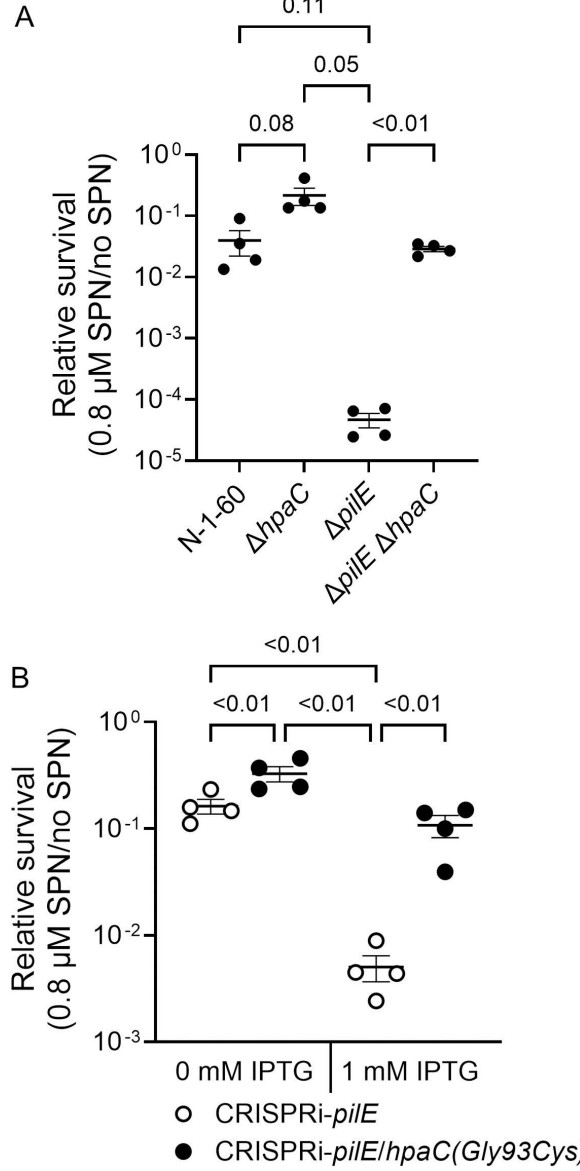

Fig 2. HpaC inhibits SPN survival in a T4p-independent manner. (A) Testing the effect of a ΔhpaC mutation in piliated and non-piliated Gc on streptonigrin sensitivity. The relative survival of FA1090 (N-1-60), ΔhpaC (N-7-37), ΔpilE (N-1-69), and ΔpilEΔhpaC (N-7-52) to SPN. (B) The effect of repressing *pilE* expression in a strain that carries *hpaC(Gly93Cys)* on SPN sensitivity. CRISPRi-*pilE* (N-5-22) and CRISPRi-*pilE/hpaC(Gly93Cys)* (N-7-15) were treated with or without 1 mM IPTG to repress *pilE* expression and with or without 0.8 µM SPN. The average relative survival of biological replicates and standard errors are shown. An ANOVA was used to determine statistical significance in A and a two-way ANOVA for B.

results indicate that HpaC inhibits SPN resistance, HpaC functions independently of the T4p, and the Gly93Cys change is a loss-of-function mutation.

## HpaC promotes LL-37 and $H_2O_2$ resistance

Since HpaC affects SPN resistance, we asked whether HpaC impacted other pilus-dependent phenotypes. We previously reported that T4p promotes resistance to LL-37 and $H_2O_2$ killing [19,25]. We deleted *hpaC* in piliated and nonpiliated Gc

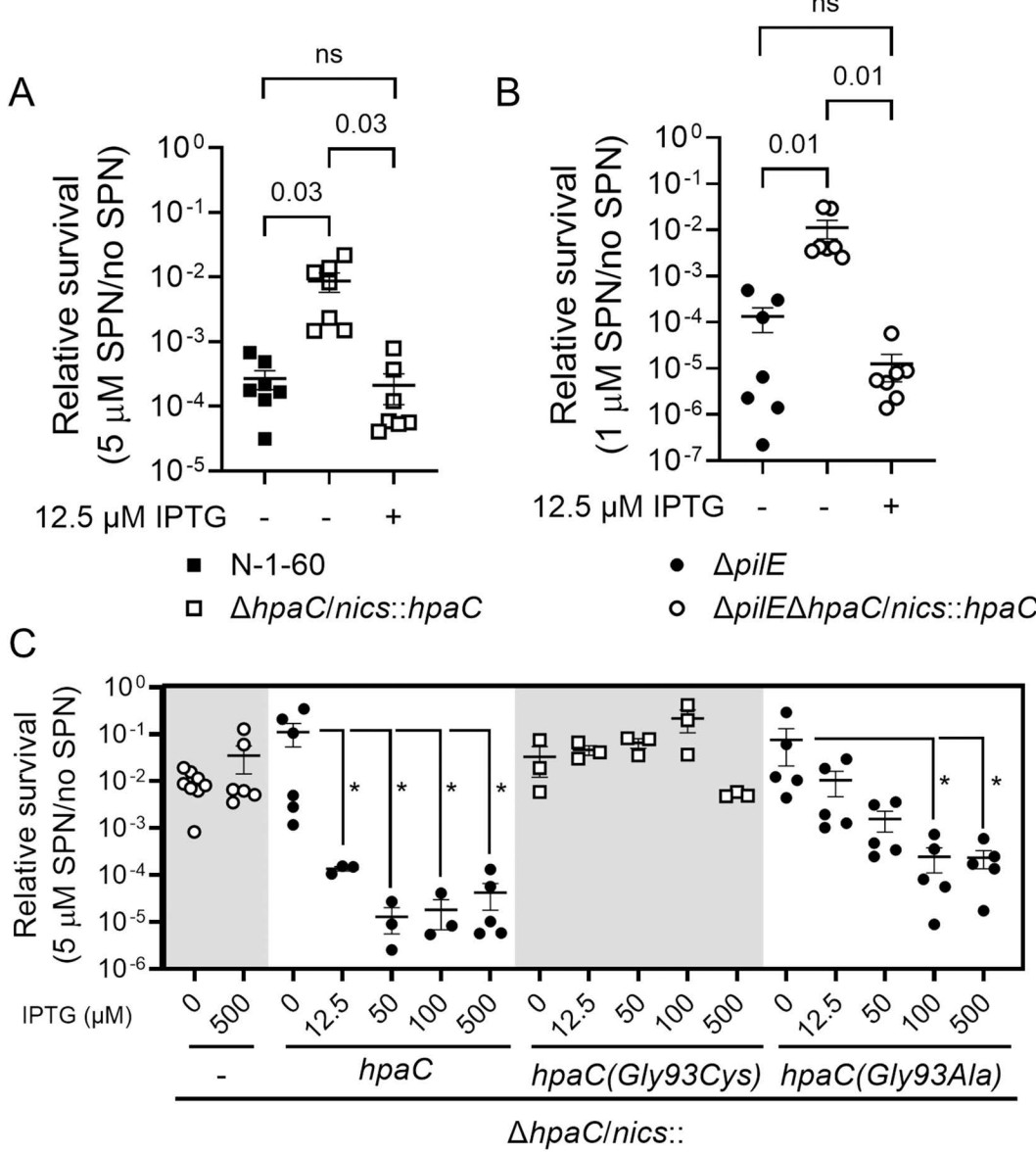

**Fig 3. ΔhpaC complementation and HpaC overexpression.** Complementing an ΔhpaC mutation in piliated and non-piliated FA1090. The effect of inducing *hpaC* expression with IPTG from the *lctP/aspC nics* region in the chromosome in (A) an ΔhpaC mutant (N-8-53) compared to the N-1-60 parent strain or in (B) a *ΔpilEΔhpaC* (N-8-63) mutant compared to a *ΔpilE* mutant on SPN sensitivity. (C) Overexpressing HpaC(Gly93Cys) has no effect on SPN sensitivity. Comparing the effect of ΔhpaC (N-8-57), ΔhpaC/nics::hpaC (N-8-53), ΔhpaC/nics::hpaC(Gly93Cys) (N-9-38), and ΔhpaC/nics::hpaC(Gly93Ala) (N-9-1) on SPN relative survival. The average relative survival and standard error of biological replicates are shown, and an ANOVA was used to determine the statistical significance.

and measured the mutants' relative survival to LL-37 and $H_2O_2$. While HpaC had no effect at lower concentrations of LL-37 or $H_2O_2$ (Fig 4, S1A, S1C Figs), at a higher LL-37 or $H_2O_2$ concentration, PilE and HpaC promoted resistance (Fig 4, S1B, and S1D). Interestingly, deleting both *hpaC* and *pilE* did not significantly affect their sensitivity to LL37 compared to the single mutants (Fig 4, S1A, S1B). While the double mutant was more sensitive to $H_2O_2$ than either single mutant (Figs 4B and

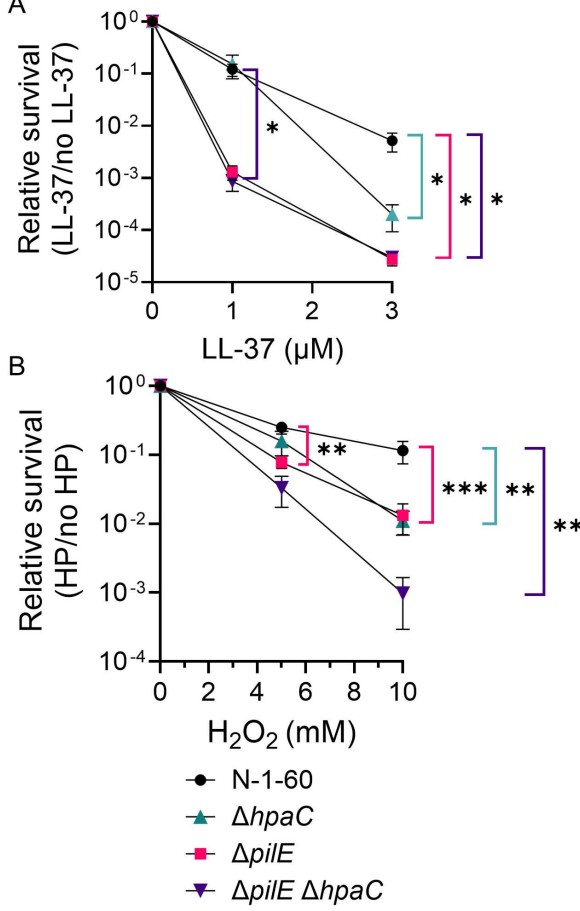

**Fig 4. HpaC promotes T4p-dependent resistance to higher concentrations of LL-37 and H₂O₂.** The relative survival of FA1090 (N-1-60), $\Delta hpaC$ (N-7-37), $\Delta pilE$ (N-1-69), and $\Delta pilE\Delta hpaC$ (N-7-52) to an LL-37 concentration gradient (A) or H₂O₂ concentration gradient (B). Ordinary one-way ANOVA was performed to determine statistical significance (* = $p < 0.05$, ** = $p < 0.01$, *** = $p < 0.001$). The standard error of the means are shown for at least five biological replicates in A and for at least nine biological replicates for each strain in B.

S1D Figs), these differences were not statistically significant (p-value >0.99). Therefore, HpaC promotes LL-37 and H₂O₂ resistance and the mechanism is in a T4p-dependent manner.

We also tested whether HpaC affects Gc sensitivity to other antimicrobials. We tested six other antibiotics that disrupt multiple cellular targets (cell wall synthesis: ampicillin, ceftazidime; cell envelope: polymyxin B; DNA replication: nalidixic acid; transcription: rifampicin; and translation: tetracycline) (S2 Fig). HpaC expression had no effect and, consistent with previous findings, piliation did not affect sensitivity to these antimicrobials [19].

## The role of oxidative killing in HpaC-mediated SPN sensitivity

SPN kills Gc by reacting with intracellular labile iron and producing DNA-damaging reactive oxygen species [19,26,31]. This killing is reduced by the addition of superoxide scavenger tiron [19,32]. We tested the role of the reactive oxygen species inhibitor in HpaC-mediated SPN sensitivity by comparing the $\Delta pilE$ mutant with and without an $hpaC$ mutation (S3 Fig). Consistent with what we observed previously, tiron partially restored resistance to SPN in a $\Delta pilE$ mutant [19]. Deleting $hpaC$ in a $\Delta pilE$ mutant phenocopied this tiron-mediated partial restoration, suggesting that the deletion of $hpaC$

resolves some but not all SPN-generated oxidants in the *pilE* mutant (S3 Fig). In a Δ*pilE*Δ*hpaC* double mutant, tiron completely restored resistance to SPN (S3 Fig). These results indicate that Δ*pilE* SPN sensitivity is due to the combination of oxidative killing and the presence of HpaC.

## HpaC Glycine-93 affects FAD affinity

To understand the effect of the Gly93Cys mutation on the HpaC protein, we determined the structural context of Gly93 by generating an AlphaFold structural model of HpaC. We aligned the peptide sequence of FA1090 HpaC to the structure of the FAD oxidoreductase TftC from *Burkholderia cepacia* (PDB 3K88 with a resolution of 2Å) that was crystallized with FAD and NADH [33] (Figs 5A and S4). TftC and HpaC are likely to be homologs with a Blast E-value of 1e-29. Gly93 is on an unstructured linker (Ala89-Glu97) between two alpha helices (Asp79-Phe88 and Glu98-Ala101) (Figs 5A and S4). This loop is one of the two conserved loops that form an FAD binding pocket in FAD oxidoreductases, the second linker being Arg58-Arg61 [33] (highlighted in S4 Fig). To determine whether HpaC interacts with FAD, we compared the level of FAD-dependent fluorescence in the purified protein with and without heat-denaturation using a probe-based assay that detects FAD [34]. We purified Gc HpaC and noticed an FAD signal in the absence of heat, indicating HpaC co-purified with FAD (Fig 5B). There was also an increase in the amount of detectable FAD following heat denaturation suggesting that HpaC can release bound FAD upon protein unfolding. We then tested the effect of Gly93Cys mutation on HpaC and FAD interaction. We determined that HpaC(Gly93Cys) bound significantly more FAD than WT HpaC (Fig 5C). Therefore, a Gly93 to Cys mutation enhances FAD affinity to HpaC.

## Presence of HpaB and HpaC homologs in bacteria is variable

HpaC is a flavin reductase that participates in 4-hydroxyphenylacetate (4HPA) metabolism [35]. 4HPA is a metabolic byproduct of aromatic amino acid and phenolic compound degradation and has been identified in human saliva [36]. HpaC is known to work with its partner protein, the monooxygenase HpaB [35,37]. HpaB receives two electrons from HpaC in the form of $FADH_2$ for *ortho*-hydroxylation of 4HPA releasing 3,4-dihydroxyphenylacetate. HpaB and HpaC structure and function have been studied in several microorganisms, including *E. coli* strain W, *Acinetobacter baumannii*, *Thermus thermophilus*, *Pseudomonas putida* and *aeruginosa*, and *Klebsiella pneumoniae* [38–49]. However, we could not identify an *hpaB* homolog in Gc or any other *Neisseria* species. We asked whether other bacterial species had an *hpaC* homolog without an *hpaB* homolog (Fig 6) using the National Center for Biotechnology Institute Gene database [50]. While nine Betaproteobacteria and 37 Gammaproteobacteria had both genes (*hpaB+hpaC+*), 15 Betaproteobacteria, including *Neisseria*, and 18 Gammaproteobacteria had *hpaC* without an *hpaB* gene (*hpaC+*). Additionally, a few genera (two Gammaproteobacteria and two Betaproteobacteria) had *hpaB* without an *hpaC* homolog (*hpaB+*). Therefore, *Neisseria* is not unique for only having HpaC and suggests that HpaC has functions independent of 4HPA metabolism in *Neisseria* and possibly other bacteria.

We studied the phylogenetic distribution of the *hpaB* and *hpaC* genes to reveal possible mechanisms of evolution (Fig 6 ). We constructed a phylogenetic tree using 16S rRNA gene sequences. In clade I, gammaproteobacterial *hpaB+hpaC+* genera dominate. However, two gammaproteobacterial genera, *Morganella* and *Proteus*, are *hpaC+* in Clade I, suggesting that *hpaB* gene loss occurred within these two organisms. Clade II consists mostly of *hpaC+* genera apart from three *hpaB+hpaC+* gammaproteobacterial genera, *Conservatibacter*, *Pasteurella,* and *Lonepinella*. Clades III and IV are mixed. Clade III consists of 11 *hpaB+hpaC+*, 16 *hpaC+*, and two *hpaB+* genera and clade IV consists of three *hpaB+hpaC+*, two *hpaC+*, and two *hpaB+* genera. This distribution of *hpaB* and *hpaC* suggests that multiple independent gene loss events occurred during evolution.

## Discussion

We previously reported that a Δ*pilE* mutant is hypersensitive to SPN, LL-37, and $H_2O_2$ compared to an isogenic, piliated strain in an iron-dependent manner [19]. Here, we show that *in vitro* evolution of a Δ*pilE* mutant to increased SPN resistance identified a non-synonymous mutation in *hpaC*, resulting in a cysteine substitution at glycine 93. Gc expressing

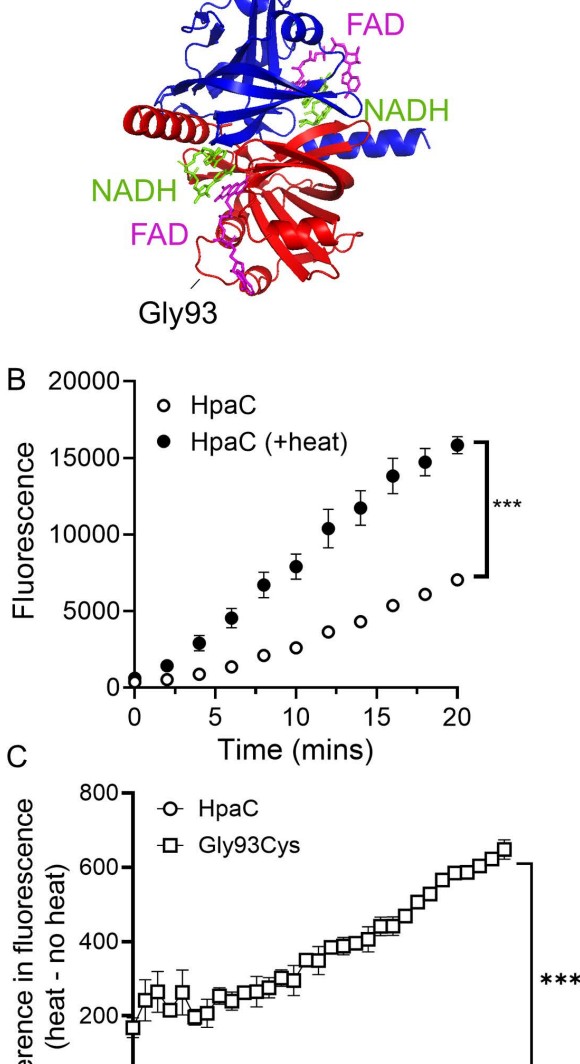

**Fig 5. HpaC(Gly93Cys) interacts with FAD better than WT HpaC.** (A) Gc HpaC structural modeling based on an alignment to *B. cepacia* TftC homod-imer (PDB ID 3K88). FAD is magenta and NADH is green. Red and blue are the two HpaC monomers. (B) FAD-dependent fluorescence assays using 0.5 µM WT HpaC, with and without heat denaturation. (C) A comparison of bound FAD fluorescence between 5 nM WT HpaC and HpaC(Gly93Cys). An Analysis of Covariance (ANCOVA) was performed to determine statistical significance of the slopes (***=p<0.001). The means and standard errors of three technical replicates are plotted.

HpaC(Gly93Cys) phenocopied a clean deletion of *hpaC* with increased SPN resistance. Thus, the point mutation is a loss-of-function mutation. An analysis of the HpaC in almost 1000 Gc isolates from PubMLST [53] showed that HpaC is highly conserved and Gly93 is present in every sequence (S4 Fig). Since an alanine substituted for Gly93 resulted in partial complementation (Fig 3C), the loss-of-function is specific to the cysteine mutation. We tested the effect of HpaC in piliated and nonpiliated Gc and determined that an Δ*hpaC* mutant increased SPN resistance irrespective of piliation. Interestingly,

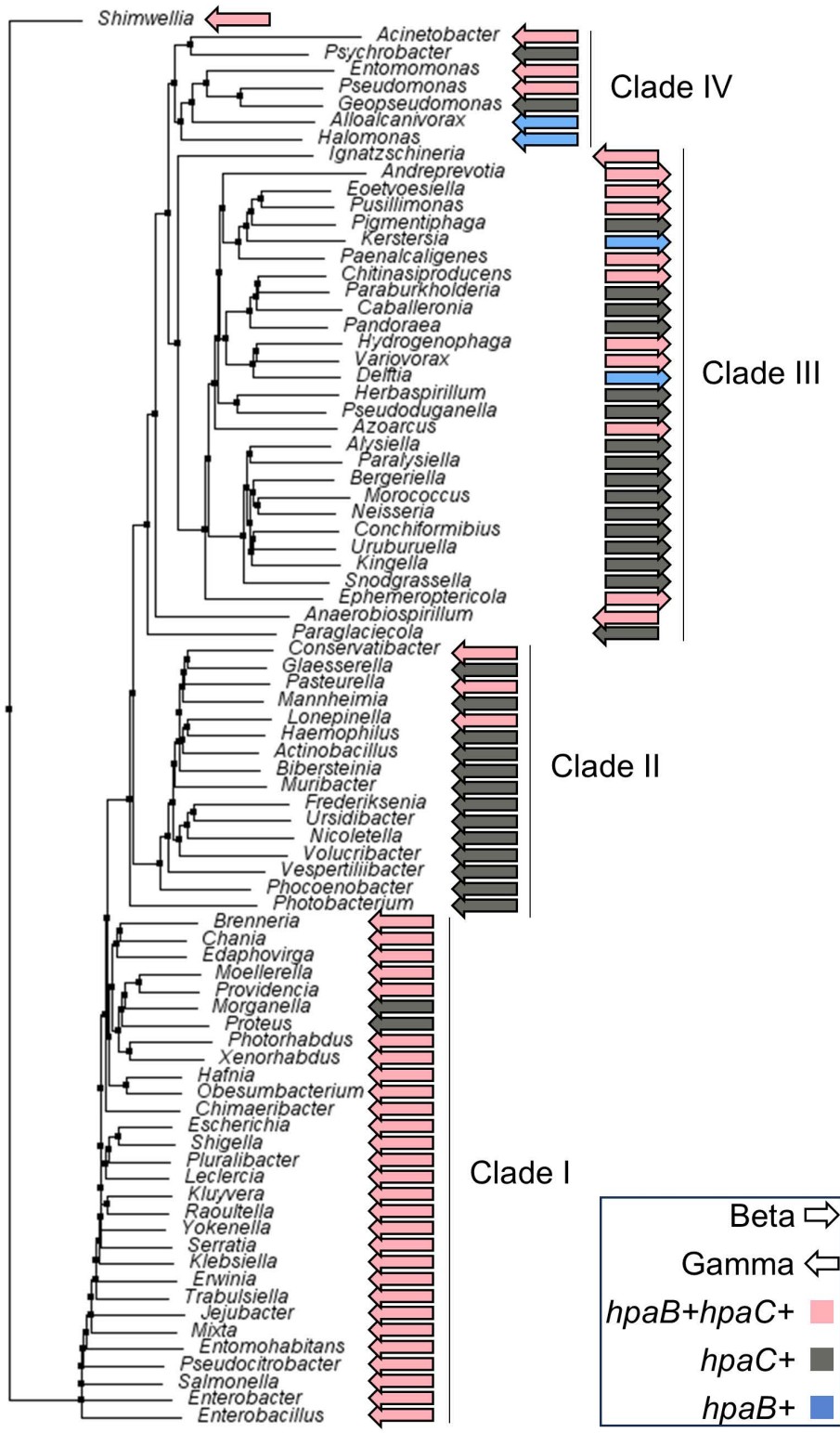

**Fig 6. The presence of *hpaB*, *hpaC*, or both in classes Betaproteobacteria and Gammaproteobacteria.** Neighbor-joining phylogenetic tree using Betaproteobacteria (right-pointing arrow) and Gammaproteobacteria (left-pointing arrow) 16S rRNA gene sequences. Pink, grey, and blue arrows indicate genera with both *hpaC* and *hpaB* (*hpaB*+*hpaC*+), *hpaC* only (*hpaC*+), or *hpaB* only (*hpaB*+), respectively. (Not all *E. coli* strains contain *hpaBC*. *E. coli* B, C, and W have both genes in an operon while *E. coli* K-12 have neither gene [51,52].

the HpaC effect was greater in the absence of pilin, suggesting that the T4p interferes with HpaC inhibition of SPN resistance. In $H_2O_2$ and LL-37 resistance, the $\Delta hpaC$ mutant exhibited increased sensitivity in piliated cells and had no effect in nonpiliated cells. Therefore, we propose that HpaC(Gly93Cys) promotes T4p-independent SPN resistance and enhances T4p-dependent $H_2O_2$ and LL-37 sensitivity in Gc (Fig 7).

Our results suggest that an HpaC(Gly93Cys) mutation reduces FAD reductase activity. Gly-93 is on a FAD binding loop and mutating this residue alters FAD affinity. We determined that FAD occupancy was higher with an HpaC(Gly93Cys) mutant than WT, indicating that mutating Gly-93 to a cysteine increases FAD affinity for HpaC. Since an HpaC(Gly93Cys) mutant suppressed SPN sensitivity, we hypothesize that the FAD:FADH$_2$ and/or NAD:NADH ratio contributes to SPN sensitivity and a glycine-to-cysteine change disrupts HpaC-dependent FAD reduction. In *P. aeruginosa*, HpaC binds FAD weakly [48]. Perhaps a low affinity for FAD is important in normal HpaC-mediated reduction. The consequence of enhanced FAD affinity with the HpaC(Gly93Cys) mutant may be an inability of HpaC to dissociate from reduced FAD. Therefore, HpaC(Gly93Cys) does not release FADH$_2$ to donate the electrons to the next cellular effector making HpaC a dead-end for FAD-dependent redox reactions.

There are multiple possible pathways through which HpaC as a flavin reductase can alter SPN, LL-37, and $H_2O_2$ sensitivity. Any one of these changes can have a broader impact on the cell. HpaC may affect the level of reactive oxygen species and, thus, sensitivity to oxidative killing. HpaC could affect the level of FADH$_2$ which can spontaneously reduce $O_2$ to $H_2O_2$ or enter the electron transport chain and, through sequential, partial reduction of $O_2$, generate superoxide, $H_2O_2$, and hydroxyl anions. Indeed, *E. coli* W HpaC can reduce cytochrome *c in vitro*, an electron carrier in the respiratory chain [45]. Alternatively, HpaC may impact a flavin-dependent metabolic pathway. Several groups have demonstrated that HpaC in other bacteria supply reduced FAD to the monooxygenase HpaB by diffusion [44,45,49] indicating that no physical interaction is needed. Therefore, reduced flavins from Gc HpaC could indirectly influence sensitivity through FADH$_2$-dependent metabolic enzyme(s).

The genomic context of *hpaC* suggests that HpaC may be a part of an antimicrobial transcriptional response. *hpaC* is adjacent to and in the same orientation as the gene *farR* which suggests they are in an operon. FarR is a transcriptional repressor of the long-chained antimicrobial fatty acids efflux pump FarAB [54,55]. FarR regulates its own expression and is under MtrR (multiple transferable resistance) transcriptional regulation [54]. MtrR also regulates the expression of the *pilMNOPQ* operon, encoding parts of the T4p secretion system [56]. Therefore, HpaC could be part of the MtrR and FarR transcriptional antimicrobial response that coordinates with piliation status.

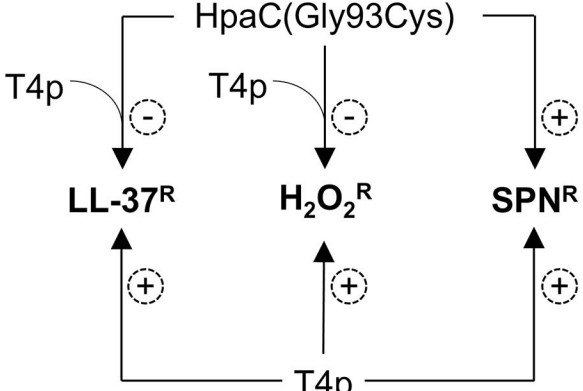

**Fig 7. Model summarizing the HpaC and T4p effects on SPN, LL-37, and $H_2O_2$ sensitivity in Gc.** The T4p is required for resistance against SPN, LL-37, and $H_2O_2$. HpaC(Gly93Cys) promotes SPN resistance in a T4p-independent manner while promoting LL-37 and $H_2O_2$ sensitivity through a mechanism that involves the T4p.

Although we have not measured the effect of HpaC on labile intracellular iron in Gc, it is unlikely that HpaC alters intracellular labile iron levels. Elevated iron in nonpiliated cells promotes SPN, LL-37, and $H_2O_2$-mediated death [19], but HpaC enhances SPN sensitivity and promotes LL-37 and $H_2O_2$ resistance. Therefore, HpaC may affect sensitivity in an iron-independent pathway.

Bioinformatic analysis of other bacterial genomes revealed that, like *Neisseria*, many bacteria lack the 4-HPA monooxygenase HpaB while encoding HpaC. The phylogenetic distribution of *hpaB* and *hpaC* suggests that these genes were present in an early common ancestor(s), and some descendants have lost either the *hpaB* or *hpaC* gene. It is possible that HpaB or HpaC function, perhaps in the metabolism of 4-HPA and similar compounds, was deleterious to fitness and, therefore, subject to negative selection. Alternatively, there may be other gene products that substitute for HpaB or HpaC that are not revealed by sequence queries. The function of HpaC in the absence of HpaB or alternative metabolic enzymes is an interesting field for further investigation.

In summary, we characterized a role for HpaC in resistance to SPN, LL-37, and $H_2O_2$ in Gc. We predict that it plays an interesting role in other bacteria that is independent of 4-HPA metabolism. To the best of our knowledge, this is the first report of the biological role of HpaC in a context that does not involve HpaB. The impact of HpaC in Gc is linked to its function as an FAD reductase and, given the critical roles that FAD redox has in the cell, HpaC is likely to affect metabolism and energy production.

## Materials and methods

### Strains and growth

Gc and *E. coli* strains and plasmids used in the study are listed in Table 1. To control for T4p phase variation, the experiments performed here used an FA1090 strain designated N-1–60 which contains point mutations in the *pilE* G4 and is *pilC2* phase-locked "on" [20]. Plasmids were introduced into Gc by spot transformation [57]. For solid medium, Gc was grown on gonococcal growth broth: 36.25 g/L GC Medium Base (Difco), 1.25 g/L agar, and Kellogg supplements I and II at 37 °C in 5% CO2. For liquid growth, 50 mM sodium bicarbonate and supplement I was added to Gc broth and cells were grown with aeration at 37 °C. Antibiotics and their concentrations used for selection in GCB were kanamycin (Kan) 50 μg/ml and erythromycin (Erm) 1 μg/ml. NEB 5-alpha competent *E. coli* (High Efficiency) cells used to propagate plasmids and NEB BL21 (DE3) Competent *E. coli* used to express protein were grown in Luria-Bertani (LB) solid medium containing 15 g/L agar or liquid media at 37 °C. The antibiotics and their concentrations used in LB were ampicillin (Amp) 100 μg/ml and Kan 50 μg/ml.

### Mutant construction

***hpaC::kan* mutant.** An approximately 650 bp fragment containing 270 bases upstream of NGO0059, the first 30 bases of NGO0059, a PacI restriction site, HA tag, NotI restriction site, the last 30 bp of NGO0059, and 270 bp downstream of NGO0059, was synthesized and cloned into pTwist-Amp-MC vector by Twist Biosciences.

A PacI and NotI flanked *nptII* fragment with a 12-mer DNA uptake sequence at the 3' end of *nptII* from a previously generated plasmid [58] was introduced into the PacI- and NotI- digested plasmid from Twist Biosciences in between the upstream and downstream sequences of NGO0059. This plasmid pTwist-*hpaC*::*kan* was used to spot transform Gc strains to generate Δ*hpaC*::*kan* strains. Transformants were selected on GCB Kan and checked by diagnostic PCR and sequencing.

**Δ*hpaC::EcoR1-EcoRV-BamH1* polylinker mutant.** We constructed the Δ*hpaC*::*EcoR1-EcoRV-BamH1* mutant by replacing the kanamycin resistance cassette in Δ*hpaC*::*kan* with a EcoR1-EcoRV-BamHI (5'-GAATTCGATATCGGATCC-3') polylinker sequence. To replace the kan cassette in pTwist-*hpaC*::*kan*, we ran site-directed mutagenesis on pTwist-*hpaC*::*kan* using primers NGO0059_ERB_F (5'-atcggatccAGGCCTTTAGACTGATATTC-3') and NGO0059_ERB_R

(5'-atcgaattcCTGCAAATCCGCCATTTTTC-3') with Q5 polymerase according to manufacturer protocol (NEB E0554). The DNA mix was treated with a kinase, T4 ligase, and DpnI mix for 5 minutes at room temperature before transformation into NEB 5-alpha Competent *E. coli* (NEB C2987H). We confirmed the sequence of the fragment from ampicillin resistant clones by Sanger sequencing with primers hpaC_500F (5'-gatgacccaattcaggcctattct-3') and hpaC_500R (5'-gtcgggcagcagggaaa-3'). The plasmid pTwist-*hpaC*::*EcoR1-EcoRV-BamH1* was spot transformed into Δ*hpaC*::*kan.* We screened transformants with Kan sensitivity by PCR and Sanger sequencing to confirm the presence of the polylinker sequence in *hpaC*, generating the Δ*hpaC*::*EcoR1-EcoRV-BamH1* mutant (N-8–57).

**hpaC(Gly93Cys) point mutant.** A 1532 bp PCR product carrying *hpaC(Gly93Cys)* was generated by using primers NGO0059upF (5'- ATGCCGTCTGAAATACAGGCAAGGGAAGCC-3') and NGO0059dnR (5'-TGAATGTCAGTCCGTTGCC-3') from an evolved Δ*pilE* mutant carrying *hpaC(Gly93Cys)* and purified using a Qiagen PCR cleanup kit. The PCR product was sent for Sanger sequencing to confirm the presence of the Gly93Cys mutation (G277T nucleotide change) and transformed into CRISPRi-*pilE* (N-5–22, [27]). We swabbed the cells from the spot into liquid GCBL for SPN selection with IPTG and spread on GCB agar plates. We screened individual colonies for *hpaC* the G-to-T mutation by PCR and Sanger sequencing.

### Complementation strains

We expressed *hpaC* from the transcriptionally silent *lctP-aspC Neisserial* intergenic complementation site (*nics*) [59]. PacI- and PmeI-flanked on the 5' and 3' ends of *hpaC* (NGO0059) was amplified by PCR using primers hpaC_PacI_F (5'-cctTTAATTAAatggcggatttgcagaaaact-3') and hpaC_PmeI_R (5'-aggGTTTAAACtcagtctaaa ggcctaaactgcc-3') from Gc. After PCR cleanup, the PCR product was cloned into pCR2.1-TOPO vector following manufacturer's protocol (Invitrogen K450002) resulting in pTOPO-*hpaC*. We sequenced *hpaC* using M13F and M13R primers. We amplified *hpaC* from pTOPO-*hpaC* with KOD polymerase (Sigma 71085), gel extracted the products, digested the purified PCR product and pGCC4 [59] with PacI and PmeI in rCutsmart buffer (New England Biolabs) for 1 hour at 37 °C, heat inactivated the enzymes for 20 minutes at 65 °C, and ligated the digested PCR product and pGCC4 for 2 hours at room temperature. We transformed the ligation into NEB 5-alpha *E. coli* (NEB C2987H) or JM109 competent cells (Promega L2005) and selected for Kan (for pGCC4 and pTOPO) resistance and sensitivity to Amp (for pTOPO). We confirmed the insert with Sanger sequencing using primers hpaC_PacI_F and aspCrev (5'-AGTGGAACGAAAACTCACGT-3'). We transformed pGCC4-*hpaC* plasmid into piliated *hpaC* null mutants and selected for erythromycin resistance, resulting in the construction of Δ*hpaC*/*nics*::*hpaC* (N-8–53). To construct Δ*hpaC*/*nics*::*hpaC(Gly93Cys)* (N-9–38), we used site-directed mutagenesis of pTOPO-*hpaC*, sequenced to confirm the presence of the mutation, cloned *hpaC(Gly93Cys)* into pGCC4, and transformed the plasmid into Δ*hpaC* (N-8–57). To construct Δ*pilE*Δ*hpaC*/*nics*::*hpaC* (N-8–63), we deleted *pilE* by screening for nonpiliated colonies after backcrossing genomic DNA from FA1090 Δ*pilE* (N-1–69) and validating by PCR using primers pilRBS (5'-GGCTTTCCCCTTTCAATTAGGAG-3') and SP3A (5'-CCGGAACGGACGA CCCCG-3') [20].

### Construction of plasmids for HpaC and Gly-93 variant protein expression

The pET28a vector carrying FA1090 *hpaC* was synthesized (Twist Bioscience) and transformed into BL21 DE3 *E. coli* cells (Table 1, N-8-3). We performed site-directed mutagenesis (NEB E0554S) on pET28a-FA1090 *hpaC* to generate pET28a-FA1090 *hpaC(Gly93Cys).* For Gly93Cys, we used primers FA1090_hpaC_F (5'-CGGGCTGACCTGCCTGT CGCCCG-3') and reverse primer (5'-GCAAAATGTTCGGCAACATCCTG-3'). We treated the reactions with a kit-provided kinase, T4 ligase, and DpnI mix for 5 minutes at room temperature before transforming 2.5 μl into NEB 5-alpha *E. coli* (NEB C2987H). After confirmation of the mutation with sequencing, we transformed BL21 DE3 cells and selected for Kan resistance.

### *In vitro* evolution of Δ*pilE* mutant and backcrossing into CRISPRi-*pilE*

The *pilE* clean deletion mutant was grown on GCB agar plates for 16 hours at 37 °C with 5% CO2. Cells were swabbed and inoculated into 15 ml conical tubes containing 1 ml GCBL with supplement I and sodium bicarbonate, shaking for 1.5 hours at 37 °C. The culture was diluted 1:5 with 5 ml GCBL with supplement I and sodium bicarbonate and grew for 1.5 hours at 37 °C. The culture was diluted to an OD550 0.2 and treated with 0.5µM SPN or DMSO for 30 minutes at 37 °C. Cells were pelleted at 4000 rpm for 2 minutes and resuspended in fresh GCBL. Cells were spread onto GCB agar plates, grown overnight at 37 °C with 5% CO2, and stored at -80 °C in GCBL and 20% glycerol and swabbed to enter the next round of growth and selection. The process of growth and selection was repeated six times and relative survival compared to the unevolved Δ*pilE* mutant was performed three times during the experiment. Genomic DNA was isolated at the sixth round of selection using phenol chloroform extraction and submitted for Illumina sequencing with SeqCenter in Pittsburgh, Pennsylvania. We aligned the reads to the reference sequence N-1–60 (Accession number CP115654.1) and called variants relative to the reference using Snippy v4.6.0 then filtered only for variants that were unique to the evolved isolates (S1 Table).

For isolating the mutations, we spot transformed the gDNA into the CRISPRi-*pilE* strain (N-5–22) [27]. The patch of cells that were exposed to the gDNA mix was swabbed into GCBL, diluted 1:5 with GCBL with supplement I and sodium bicarbonate and grew for 1.5 hours at 37 °C, cultures were diluted to an OD550 0.05, and grew with and without 1 mM IPTG for 2 hours before treatment with 0.5 µM SPN or DMSO for 30 minutes at 37 °C. Cells were washed by pelleting at 4000 rpm for 2 minutes and resuspending in fresh GCBL. We performed 10-fold dilutions of the cultures onto GCB agar plates and overnight at 37 °C with 5% $CO_2$. Individual colonies were frozen in 100 µl of GCBL with glycerol in 96 well plates. Isolates were tested for SPN sensitivity compared to CRISPRi-*pilE* in the presence and absence of IPTG. Genomic DNA prepared with phenol chloroform were sent for whole genome sequencing and variant calling with SeqCenter. Sample libraries were prepared using the Illumina DNA Prep kit and IDT 10 bp UDI indices, and sequenced on an Illumina NextSeq 2000, producing 2x151bp reads. Demultiplexing, quality control and adapter trimming was performed with bcl-convert (v3.9.3), a proprietary Illumina software for the conversion of bcl files to basecalls. Variant calling was carried out using Breseq under default settings [60].

### Gc antimicrobial killing assays

Cells were grown on GCB agar plates for 16–18 hours at 37 °C with 5% CO2. Individual colonies representing biological replicates were picked and streaked onto GCB agar plates and incubated for 16–18 hours at 37 °C with 5% CO2. Cells were inoculated to an OD550 between 0.2-0.3 in 1 ml GCBL with supplement I and sodium bicarbonate and grew for 1.5 hours at 37 °C. Cultures were diluted 1:5 with 5 ml GCBL with supplement I and sodium bicarbonate and grew for 1.5 at 37 °C. Cultures were diluted to an OD550 0.05 and grown with and without 1 mM IPTG if the strain carries a CRISPR interference array for 1.5-2 hours. The conditions for hydrogen peroxide, LL-37, and SPN killing assays and the method to determine the relative survival then follow the previously described protocol [19].

### Complementation

For complementation, we grew Gc to mid-exponential phase, normalized the cultures to an OD550~0.1, and treated the cultures with or without 12.5 µM IPTG for 1 hour at 37 °C. We then treated piliated FA1090 and Δ*hpaC*/*nics*::*hpaC* (N-1–60 and N-8–53) with 5 µM SPN and nonpiliated Δ*pilE* and Δ*pilE* Δ*hpaC*/*nics*::*hpaC* (N-1–69 and N-8–63) 1 µM SPN for 20 minutes at 37 °C.

### Purification of FA1090 HpaC and HpaC(Gly93Cys)

A 2.5 ml overnight culture of BL21 DE3/pET28a-FA1090 *hpaC* or BL21 DE3/pET28a-FA1090 *hpaC(Gly93Cys)* mutant was diluted into 250 ml LB with kanamycin in a 1 L flask and grew for 2 hours shaking at 37°C. IPTG was added to a

final concentration of 0.1 mM and grew overnight with shaking at 37 °C. Cells were pelleted at 4500 rpm at 4°C. The cell pellets were resuspended in 10 ml resuspension buffer (50 mM Tris-HCl, pH 8, 2 mM ethylenediaminetetraacetic acid). Cells were centrifuged and frozen at -80 °C overnight. To lyse the cells, the pellets were resuspended in 25 ml lysis buffer (50 mM NaH2PO4, pH 8, 0.5 M NaCl, 1% triton X-100, with 2 tablets of freshly added cOmplete Mini protease inhibitor, Roche 11836153001) and pulse sonicated for 5 minutes at 35% amplitude in 30 second intervals separated by 20 seconds on ice. Lysates were centrifuged at 4500 rpm at 4°C for 30 minutes. The supernatants were treated with DNase (300 units/25 ml lysate), incubated for 1 hour at room temperature, and passed through a 0.45 µm filter and stored at 4°C. We equilibrated 2.5 ml of cOmplete His-Tag Purification Resin (Roche 5893682001) with 20 column volumes (50 ml) of buffer A (50 mM NaH2PO4, pH 8, 300 mM NaCl). We added the equilibrated resin to the cell lysates and mixed gently overnight at 4°C. We packed the resin-lysate mix by gravity-assisted flow onto a column and the flow-through was collected. We washed the resin with buffer A with 20 mM imidazole and collected the washes in separate tubes for later analysis. We eluted the His-tagged protein with buffer A with 70 mM imidazole into separate fractions and stored the samples at 4°C. We ran the flow-through, washes and elutions on an SDS-PA gel to check the quality of the purification. The most concentrated elution fractions were pooled, diluted to 1 mg/ml, and dialyzed with protein storage buffer (50 mM Tris-HCl, pH 8, 150 mM NaCl, 50% v/v glycerol) overnight at 4°C and stored at -80 °C. We determined the concentration of protein using a BCA kit. We purified 51 µM HpaC and 15 nM HpaC(Gly93Cys).

### FAD assays

We used a kit that uses FAD as a cofactor for an oxidase enzyme mix, generating a product that a probe detects, resulting in either fluorescence or a change in color (Abcam ab204710). We used the kit according to manufacturer provided protocol except at room temperature and with half of the recommended OxiRed probe and oxidase enzyme to conserve reagents. Purified His6-tagged FA1090 HpaC or HpaC(Gly93Cys) was denatured by heating at 95 °C for 5 minutes or not, cooled on ice, and added to the FAD assay reactions in black polystyrene 96-well assay plates (costar 3915). Fluorescence was measured every two minutes with excitation and emission maxima of 535/587 nm.

### Aligning FA1090 HpaC to *B. cepacia* TftC

We aligned the peptide sequence of FA1090 HpaC to TftC from *B. cepacia* (PDB 3K88). We used the "align" command in PyMOL with five refinement cycles for outlier rejection. The analysis yielded an alignment with a root mean square deviation (RMSD) 0.684 Å for the backbone $C_\alpha$ atoms.

### Bioinformatic analysis of bacterial genomes containing *hpaB* and/or *hpaC*

We searched for the terms "*hpaC*" and "*hpaB*" on the National Center for Biotechnology Institute Gene database [50], accessed each genus listed in the taxonomic groups of Betaproteobacteria (26 genera) and Gammaproteobacteria (57 genera), confirmed the presence of HpaC and HpaB by protein sequence homology, and analyzed the genomic context surrounding either *hpaC* or *hpaB* (~10 genes up or downstream) to identify if the other gene was present. We manually curated the genera into categories of genomes having either *hpaC*, *hpaB*, or both. After ClustalW alignment of the 16S rRNA sequences, we generated neighbor-joining rooted phylogenetic trees in Jalview.

## Supporting information

**S1 Fig. Concentration-dependent HpaC involvement in LL-37 and hydrogen peroxide resistance.** Relative survival of FA1090 (N-1–60), Δ*hpaC*::*ERB* (N-8–59), Δ*pilE* (N-1–69), and Δ*pilE*Δ*hpaC* (N-8–61) to LL-37 (A and B) and hydrogen peroxide (C and D). A one-way ANOVA was used to determine statistical significance. The means and SEMs are plotted for biological replicates.
(TIF)

**S2 Fig. HpaC has no significant effect on other antibiotic sensitivities.** The minimum inhibitory concentrations for six antimicrobials were determined using E-test strips (µg/ml). Three biological replicates of the parental strain (N-1–60) and isogenic mutants Δ*pilE* (N-1–69), Δ*hpaC* (N-8–59), and a double mutant Δ*pilE*Δ*hpaC* (N-8–61) were tested. Statistical significance was determined by a one-way ANOVA followed by a Sidak's multiple comparisons test.
(TIF)

**S3 Fig. The effect of antioxidant tiron (0.5 mM) on streptonigrin (1 µM) sensitivity in the absence of PilE (Δ*pilE*, N-1–69) or without both PilE and HpaC (Δ*pilE* Δ*hpaC*, N-7–52).** Average relative survival is shown with standard error of the mean for six biological replicates. A 2way ANOVA followed by a Dunnett's multiple comparisons test was used to determine statistical significance.
(TIF)

**S4 Fig. Structural modeling of FA1090 HpaC.** A) Overlay of *B. cepacia* TftC (PDB 3K88 in cyan) and FA1090 HpaC (red). Glycine-93 is highlighted in orange. B-D) FA1090 HpaC modeled with FAD (purple) and NADH (green) and the two conserved FAD binding loops (white). HpaC from B is rotated 90° to the right (C) or 90° up (D) to show different perspectives of the FAD binding pocket. E) *N. gonorrhoeae* HpaC consensus logo sequence. A consensus sequence of 999 NEIS0375 HpaC homologs with a peptide sequence of 166 residues from PubMLST using weblogo.berkley.edu. Underlined are the two conserved FAD binding loops. Marked with an asterisk is amino acid residue 93.
(TIF)

**S1 Table. Genetic differences in streptonigrin resistant Δ*pilE* mutant.** After *in vitro* evolution, reads from whole genome sequencing were aligned to an unevolved reference strain N-1–60 [19] and variants unique to the evolved Δ*pilE* mutant were called.
(PDF)

## Acknowledgments

The authors thank the current and past members of the Seifert lab for their insight and support. In particular, we thank Drs. Selma Metaane for her valuable feedback of the manuscript and Wendy Geslewitz for her thoughtful conversations on this work.

## Author contributions

**Conceptualization:** Linda I. Hu, H S. Seifert.

**Data curation:** Linda I. Hu, Egon A. Ozer.

**Formal analysis:** Linda I. Hu, Egon A. Ozer.

**Funding acquisition:** H S. Seifert.

**Investigation:** Linda I. Hu.

**Methodology:** Linda I. Hu.

**Project administration:** H S. Seifert.

**Software:** Egon A. Ozer.

**Supervision:** H S. Seifert.

**Validation:** Linda I. Hu.

**Visualization:** Linda I. Hu.

**Writing – original draft:** Linda I. Hu.

**Writing – review & editing:** Linda I. Hu, Egon A. Ozer, H S. Seifert.

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
