## [Decision Letter · Decision Letter 0]

22 Aug 2025

PPATHOGENS-D-25-01885

The flavin reductase HpaC differentially sensitizes *Neisseria gonorrhoeae* during Type IV pilus-dependent killing

PLOS Pathogens

Dear Dr Hu,

Thank you for submitting your manuscript to PLOS Pathogens. Your manuscript has been evaluated by members of the editorial board and three external referees. All three reviewers expressed positive comments about the manuscript. They do, however, have comments that need attention with appropriate editorial changes (see below). Note that additional experiments are not required. Upon receipt of the revised manuscript with an accompanying response to each comment a final decision will be possible. Therefore, we invite you to submit a revised version of the manuscript that addresses the points raised during the review process.

Please submit your revised manuscript within 30 days Oct 21 2025 11:59PM. If you will need more time than this to complete your revisions, please reply to this message or contact the journal office at plospathogens@plos.org. Please include the following items when submitting your revised manuscript:

We look forward to receiving your revised manuscript.

Kind regards,

William M Shafer, Ph.D.

Guest Editor

PLOS Pathogens

D. Scott Samuels

Section Editor

PLOS Pathogens

Sumita Bhaduri-McIntosh

Editor-in-Chief

PLOS Pathogens

orcid.org/0000-0003-2946-9497

Michael Malim

Editor-in-Chief

PLOS Pathogens

orcid.org/0000-0002-7699-2064

**Journal Requirements:**

2) Please ensure that the funders and grant numbers match between the Financial Disclosure field and the Funding Information tab in your submission form. Note that the funders must be provided in the same order in both places as well.

State what role the funders took in the study. If the funders had no role in your study, please state: "The funders had no role in study design, data collection and analysis, decision to publish, or preparation of the manuscript.".

**Reviewers' Comments:**

Reviewer's Responses to Questions

**Part I - Summary**

Reviewer #1: The strength of the study was use of superior genetic technology iCRISP to induce or suppress gene expression in concert with other mutagenesis technology to thoroughly characterise the role of HpaC in resistance phenotypes towards oxidative stress and antimicrobial peptides.

The novelty was derived from finding the involvement of a flavin reductase in antibiotic resistance in Neisseria.

The weaknesses were minor. I think that the model presented in Figure 5 could be improved as it is known that mtrR regulates pili expression and farA and could be incorporated.

This was a well written and cited article and raised no concerns for me.

Reviewer #2: This is an interesting paper describing the findings of an in vitro evolution experiment looking for the basis of SPN sensitivity in the goncococus (Gc). The authors identify a SNP in HpaC which they show convincingly affect SPN sensitivity in a T4p-independent manner. Interestingly the authors show that Gc have an orphan HpaC (i.e. present without HpaB), which is a previously unappreciate feature of other bacteria. The SNP phenocopies a HpaC null mutant in its sensitivity to antimicrobials and the role of T4p. Quite how this occurs is not clear, although they provide evidence that the SNP enhances the interaction between FAD and the recombinant protein. The mechanism of action of HpaC's invovlement in resistant is not defined in this paper, and I think it is beyond the scope of the current manuscript.

The Figures are well presented and findings justify the conclusions. The paper is well written but there are areas that would benefit from clarification.

Reviewer #3: Hu et al. describe the in vitro evolution of N. gonorrhoeae pilE mutants to identify loci involved in streptonigrin sensitivity. A mutation was found in hapC, encoding an FAD reductase. pilE and hpaC mutants and double mutants were analyzed for effects on hydrogen peroxide-, antibiotic-, and LL-37 resistance. The results add to our understanding of the ways that type IV pili are involved in providing hydrogen peroxide and antimicrobial peptide resistance.

The description of the identification of the hpaC mutation and the analyses of the mutants is straightforward and readily understandable. That is helpful to the reader for understanding this complex system.

**Part II – Major Issues: Key Experiments Required for Acceptance**

Reviewer #1: In the discussion (line 317-322), the authors postulate that the FAD reductase HpaC could be co-expressed with the known antibiotic resistance regulator, FarR, in an operon. FarR itself is regulated by the antibiotic resistance regulator MtrR which has also been shown to regulate pili expression (J Bacteriol. 2007 May 4;189(13):4569–4577). I feel that if the authors could show that HpaC and FarR are in an operon, they can improve the sophistication of their model in Figure 5 by incorporating the known regulatory pathway of mtrR controlling both pili expression and HpaC. I feel this would improve the impact of the paper,

Reviewer #2: I dont have any experiments that need doing.

However, it would be interesting to search Gc genome databases to examine naturally occurring polymorphisms in HpaC, and see if any map to similar sites or might be predicted to give a loss of function.

Reviewer #3: No experiments required.

It is not quite convincing that the hpaC pilE double mutant is not different from the pilE and hpaC single mutants for H2O2 resistance (Fig 4B). The authors should tone down their claims of no effect in the double mutant in lines 193 and 284.

**Part III – Minor Issues: Editorial and Data Presentation Modifications**

Reviewer #1: In Figure 1 panel C, I think the last two panels are mislabelled as T1 and T2 and are meant to be T3 and T4.

Table 3 and 4: I do prefer to see this information in phylogenetic trees as they are usually more visually memorable than a Table. This is not an essential suggestion but it could look better in the final publication.

FAD reductases have known roles in antibiotic resistance, the most well known of these being the flavin reductases involved in metronidazole sensitivity. To help with the global impact of the observation made here I would have liked to see some mention of this as it would draw readers from other areas and not just Neisseria.

Reviewer #2: I suggest that the authors are more careful with the title, and also the summary figure. The authors do not demonstrate that HpaC is a flavin reductase. They might think this is the case, and could strengthen this by more careful comparison of its predicted structure with characterized related enzymes. However in the absence of biochemical data, they shouldnt state this in the title based on the evidence presented in the article, and the summary figure should include a ?mark.

I think T3 and T4 are incorrectly labeled in Fig 1C as T1 and T2 (again). Also change 'treated with SPRN' to 'backcrossed with SPNR' as it sounds like the gDNA was in the assay. And based on genome sequencing are T3 and T4 identical? If so, say so.

The discussion/analysis of the Alphafold models should be improved. What are the active site residues in this class of enzymes? Are they conserved in Gc? These sies should be shown zoomed in. They should overlay the known and predicted structures to give confidence to the reader of how closely related they are. How often were simulations run? Where is the substitution in relation to FAD? A single view of the structure is not helpful.

Authors should be careful re the use of the term binding affinities or binding when affinities/interaction alone would be sufficient as it includes off as well as on rates during the association of the protein with its substrate. For example, the substitution could trap the product (assumed to be FADH2) or indeed limit catalysis so increase association with the substrate. Neither of these mechanisms affect binding per se.

The authors should discuss why the screen did not identify HpaC loss of function nonsense mutants. These should be much more common than a non-synonymous SNP that alters protein function.

Tables 3/4 The parameters of searches should be described in detail (ie protein or nucleotide, cutoffs for identity/similarity and for %coverage, whether they were reiterative, of what databases specifically etc). In this way, it will be clear to the reader whether the absence of HpaB is correct or is due to searches performed with excessive stringency.

Minor comments:

l61 opacity proteins

l69 which neutrophil membrane

l134/Fig 1C I dont think it is correct that the 'two transformants... were more resistant to SPN when we repressed pilE than without repression'. The relative surivival of both mutants went down on repression of pilE. This sentence could be split in two.

Reviewer #3: 1. Line 52. Delete the word “main”. Gc is the only cause of “gonorrhea”.

2. Line 53. Correct to “second leading bacterial sexually transmitted infection…”. Viral STIs are more common.

3. Line 59. While the sentence about macrophage PRRs responding to Gc peptidoglycan and OM components is correct, it leaves the impression that those are the only cells responding. Epithelial cells also respond to those components and call for the neutrophil infiltration.

4. Line 78. It is the pilus that is assembled and disassembled, whereas this sentence makes it sound like the PilE protein is being taken apart. Reword.

5. Lines 100-101. It is 1/100 to 1/1000 pilE Gc that survive SPN, not that many that are killed. Similarly, it is 1/10 WT Gc that survive SPN.

6. Line 110. It seems odd that adjacent genes lgtC and lgtD would have such different NGO numbers, since the genes were numbered by chromosomal location. Can NGO11620 really be the gene next to NGO2158. Check the numbers.

7. Lines 134-136. This sentence is ambiguous in terms of what two values are being compared (pilE repressed vs unrepressed or resistance between mutants). I know they meant the latter, but the sentence is confusing and imprecise.

8. Line 148. Spell “pilE”.

9. Line 178. Since the authors used IPTG to repress genes (through CRISPRi) in the previous figures, it would be helpful to indicate in the legend or the figure (Fig. 3) that IPTG is being used to induce expression of the hpaC complement in this experiment.

10. The label in Fig. 1C is probably meant to read “T1, T2, T3, T4”.

11. The legend for the model would be more helpful if it was more descriptive of the figure and the conclusions of the study. Also, what are the blue dots supposed to represent?

PLOS authors have the option to publish the peer review history of their article (what does this mean? ). If published, this will include your full peer review and any attached files.

**Do you want your identity to be public for this peer review?** For information about this choice, including consent withdrawal, please see our Privacy Policy .

Reviewer #1: No

Reviewer #2: No

Reviewer #3: No

**Figure resubmission:**
---

## [Editor Report · Decision Letter 1]

9 Oct 2025

Dear Dr. Hu,

We are pleased to inform you that your manuscript 'Flavin affinity for the reductase HpaC differentially sensitizes *Neisseria gonorrhoeae* during Type IV pilus-dependent killing' has been provisionally accepted for publication in PLOS Pathogens.

Best regards,

William M Shafer, Ph.D.

Guest Editor

PLOS Pathogens

D. Scott Samuels

Section Editor

PLOS Pathogens

Sumita Bhaduri-McIntosh

Editor-in-Chief

PLOS Pathogens

orcid.org/0000-0003-2946-9497

Michael Malim

Editor-in-Chief

PLOS Pathogens

orcid.org/0000-0002-7699-2064

---

## [Editor Report · Acceptance letter]

Dear Dr. Hu,

We are delighted to inform you that your manuscript, "Flavin affinity for the reductase HpaC differentially sensitizes *Neisseria gonorrhoeae* during Type IV pilus-dependent killing," has been formally accepted for publication in PLOS Pathogens.

Best regards,

Sumita Bhaduri-McIntosh

Editor-in-Chief

PLOS Pathogens

orcid.org/0000-0003-2946-9497

Michael Malim

Editor-in-Chief

PLOS Pathogens

orcid.org/0000-0002-7699-2064